# S-Nitrosylation in Tumor Microenvironment

**DOI:** 10.3390/ijms22094600

**Published:** 2021-04-27

**Authors:** Vandana Sharma, Veani Fernando, Joshua Letson, Yashna Walia, Xunzhen Zheng, Daniel Fackelman, Saori Furuta

**Affiliations:** Department of Cancer Biology, University of Toledo Health Science Campus, 3000 Arlington Ave., Toledo, OH 43614, USA; vandana.sharma@utoledo.edu (V.S.); VeaniRoshale.Fernando@rockets.utoledo.edu (V.F.); Joshua.Letson@rockets.utoledo.edu (J.L.); Yashna.Walia@rockets.utoledo.edu (Y.W.); Xunzhen.Zheng@utoledo.edu (X.Z.); Daniel.Fackelman@utoledo.edu (D.F.)

**Keywords:** NO, S-nitrosylation, NOS, microenvironment, tumor-associated immune cells, microbiome, cancer therapeutics, ECM

## Abstract

S-nitrosylation is a selective and reversible post-translational modification of protein thiols by nitric oxide (NO), which is a bioactive signaling molecule, to exert a variety of effects. These effects include the modulation of protein conformation, activity, stability, and protein-protein interactions. S-nitrosylation plays a central role in propagating NO signals within a cell, tissue, and tissue microenvironment, as the nitrosyl moiety can rapidly be transferred from one protein to another upon contact. This modification has also been reported to confer either tumor-suppressing or tumor-promoting effects and is portrayed as a process involved in every stage of cancer progression. In particular, S-nitrosylation has recently been found as an essential regulator of the tumor microenvironment (TME), the environment around a tumor governing the disease pathogenesis. This review aims to outline the effects of S-nitrosylation on different resident cells in the TME and the diverse outcomes in a context-dependent manner. Furthermore, we will discuss the therapeutic potentials of modulating S-nitrosylation levels in tumors.

## 1. Introduction

S-nitrosylation, a protein modification mediated by nitric oxide (NO), exerts a myriad of biological and biochemical functions. In their pioneering study in 1992, Stamler et al. proposed that the formation of NO-adducts at protein thiols would be more stable than the gaseous NO by itself and serve as the primary mechanism for diverse NO bioactivities [1]. Since then, we have gained better understanding of the S-nitrosylation-mediated regulation of cellular signaling. However, this protein modification is less characterized at the molecular level than other modifications, such as phosphorylation. Additionally, the reported consequences of S-nitrosylation in cancer are often conflicting and are largely context-dependent. Thus, there remains a need to clearly define the role of S-nitrosylation of a particular protein in a distinct cell type and microenvironment.

Proteins with nearly all biological functions are subjected to S-nitrosylation. To date, more than 3000 proteins are found to be S-nitrosylated to modulate their conformation, function, stability, and protein-protein interaction [2,3]. Regulation of S-nitrosylation is also essential for maintaining normal or pathological cell signaling [4,5,6,7,8,9]. An imbalance in the regulation of S-nitrosylation could lead to the development of different diseases, including cancer, sepsis, and multi-organ dysfunction [10,11,12,13,14,15]. Especially, S-nitrosylation plays a critical role in the redox regulation of cells and tissues, and its dysregulation is closely linked to pathological conditions (see a comprehensive review by Fernando et al. [16]).

Cancer cells usually succumb to dysregulated levels of NO and S-nitrosylation (either hyper-S-nitrosylation or hypo-S-nitrosylation) owing to the altered expression of nitric oxide synthases (NOS) or denitrosylases as well as oncogenic mutations of target proteins. Such aberrant S-nitrosylation contributes to malignant phenotypes, such as genomic instability, cell proliferation, anti-apoptosis, angiogenesis, and metabolic reprogramming [13,17,18,19,20]. Several anti-cancer strategies are aimed at bringing elevated S-nitrosylation levels down to physiological levels to suppress the pro-tumor effects of S-nitrosylation with NOS inhibitors, NO scavengers, or denitrosylase mimetics [21,22,23,24,25,26,27]. In contrast, several other strategies are aimed to increase S-nitrosylation levels to inhibit cancer cell proliferation and promote cancer cell death and immuno-surveillance [21,28,29,30].

Furthermore, recent studies have attested to the role of S-nitrosylation as a major regulator of the tumor micro-environment (TME) [31,32,33]. TME is a complex collection of diverse populations of stromal cells, including immune cells (myeloid cells and lymphocytes) and fibroblasts as well as the extracellular matrix (ECM) [34]. Fibroblasts in the TME are activated to become cancer-associated fibroblasts (CAFs) under the paracrine signaling from tumor cells [35]. The function of tumor-associated immune cells is also modulated by tumor cells, resulting in an immuno-suppressive milieu to promote tumor progression [36]. The cellular and biochemical composition of the TME play major roles in fostering tumor cell proliferation and metastasis [37] and the refractoriness to cancer therapies. It has been increasingly evident that such tumor-promoting functions of the TME are critically regulated by NO and S-nitrosylation [32,33,38]. For example, in tumor-associated immune cells, the production of chemokines and cytokines as well as cell survival are greatly influenced by S-nitrosylation [27,38].

In this review, we will provide an overview of the role of S-nitrosylation in different cancer types, in different components of the TME, and their effects on cancer progression or suppression. We will also introduce some of the anti-cancer strategies based on modulating the levels of S-nitrosylation.

## 2. Nitric Oxide (NO) Signaling

NO is a highly reactive molecule with a half-life of 0.09~2 s [39]. NO is produced by nitric oxide synthase (NOS) using amino acid L-arginine as the substrate and a series of redox-active cofactors. In mammalian cells, there are three major NOS isoforms that encompass ~50% homology: neuronal NOS (nNOS/NOS1), inducible NOS (iNOS/NOS2), and endothelial NOS (eNOS/NOS3) [16,40]. nNOS and eNOS are expressed constitutively to produce steady-state NO levels, while their activities are regulated post-translationally, such as by phosphorylation, protein interaction, and cofactor/substrate availability [41,42]. Conversely, the expression of iNOS is regulated inducibly to produce a large amount of NO [41,42]. In addition, mitochondria are reported to possess mitochondrial NOS (mtNOS, nNOS homologue) in the matrix and inner membrane to regulate oxygen consumption and biogenesis of mitochondria [43,44,45,46,47].

The NOS monomer is composed of the carboxyl-terminal reductase and amino-terminal oxygenase domains. The functional NOS is a dimer of two identical monomers tethered by a zinc ion at two CysXXXXCys motifs, at which the substrate L-arginine and cofactor tetrahydrobiopterin (BH_4_) bind the enzyme [42,48]. In particular, BH_4_ binding allows “coupling” of the reductase and oxygenase domains for the canonical enzymatic function [40,42]. However, the reduced availability of BH_4_ or the substrate L-arginine could “uncouple” NOS, impairing the dimerization and NO production. BH_4_ deficiency could be caused by its excessive degradation under oxidative stress, which contributes to tissue fibrosis and stiffening in chronic disorders, such as obesity, cardiovascular disease, and cancer [48,49,50,51,52,53,54,55].

Most NO studies have focused on its roles in specialized cells, namely, neurons, muscles, endothelia, and immune cells. However, NO, in fact, exerts pleiotropic functions in many different types of cells, including the regulation of epithelial tissue morphogenesis, polarity formation, cellular growth, and movement [56,57,58,59,60,61,62]. NO signaling is classified into the classical and non-classical schemes. In the classical scheme, NO signaling is mediated through its binding and activation of soluble guanylyl cyclase (sGC) to convert guanosine-5′-triphosphate (GTP) to cyclic guanosine monophosphate (cGMP). cGMP, in turn, activates cGMP-dependent protein kinase (PKG) to lower the levels of potassium and calcium ions in the cytosol, leading to membrane hyperpolarization, neurotransmission, and vasodilation [41,42]. In contrast, the non-classical scheme of NO signaling includes covalent post-translational modifications of biomolecules by NO—S-nitrosylation and metal nitrosylation [63].

NO’s bioactivities are largely dependent on its concentration, timing, and context [64,65,66]. In healthy tissues, NO production is tightly regulated to attain the right condition [67]. Conversely, in a diseased state such as cancer, NO production is often dysregulated, leading to too much or too little NO levels that contribute to the disease pathogenesis [65,68,69,70,71,72]. Furthermore, NO’s activities in cancer are complex and contradictory [73]. NO could exert dichotomous effects on diverse cellular activities such as proliferation, apoptosis, angiogenesis, migration, and invasion, depending on its concentration and context [18,65,73,74,75]. For example, at lower concentrations (<300 nM), NO activates pro-tumoral signals (extracellular signal-regulated kinase [ERK] and hypoxia-inducible factor 1α [HIF1-α]), while, at higher concentrations (>300 nM), it activates anti-tumoral signals (p53) [75]. Furthermore, NO could be produced by either tumor cells or tumor-associated macrophages (M1 type), leading to either pro-tumoral or anti-tumoral effects, respectively [73,76]. Such complex NO signaling in cancer have led to conflicting reports and a notion that NO plays a double-edged role as both a cancer-promoter and cancer-inhibitor [67,68,77]. The paradoxical roles of NO in cancer would be partly resolved by clarifying a particular activity of NO in a specific stage and type of cancer under a set context.

## 3. What Is S-Nitrosylation?

The thiol group of cysteine residue is targeted for various types of covalent/post-translational modifications (PTM). These modifications play key roles in regulating protein function, subcellular localization, and protein-protein interactions [77,78]. One such modification is S-nitrosylation, which is a reversible PTM of a cysteine residue. This modification occurs via the covalent attachment of the nitrosyl (NO-) group to the thiol side chain of a cysteine, forming a S-nitrosothiol (SNO) (Figure 1) [79]. In mammalian cells, S-nitrosylation is a spontaneous reaction mediated by NO produced by NO synthases (NOS). The degrees of S-nitrosylation could range from mono-(single cysteine) to poly-S-nitrosylation (multiple cysteines), depending on the availability of NO, as well as the biochemical properties of the target proteins [3]. There are at least three determinants for the selectivity of S-nitrosylation. The first determinant is the proximity of specific cysteine residues to the NO-donor [80]. In fact, NOS and its interacting proteins are the first to become S-nitrosylated [81,82,83]. The second determinant is the presence of a signature motif (I/L-X-C-X2-D/E) that harbors a cysteine residue flanked by acidic and basic residues [15,84]. The third determinant is the presence of cysteine residues in a hydrophobic pocket that could efficiently bind NO and oxygen [10].

S-nitrosothiol-signals could then be transmitted to proteins in distant locales through transnitrosylation [85]. Transnitrosylation is a reaction that involves successive transfer of the NO-group from an already nitrosylated protein (donor) to another protein (acceptor) when the two directly interact and have the appropriate redox potential and signature motifs (Figure 1) [15,86]. During transnitrosylation, the charged amino acids in the signature motifs facilitate electrostatic protein-protein interactions. Then, the donor protein (S-nitrosylase)—with higher redox potential—passes the NO-group to the acceptor protein, while getting denitrosylated. About ten S-nitrosylases have, so far, been identified, such as S-nitrosoglutathione (GSNO), hemoglobin, thioredoxin, glyceraldehyde 3-phosphate dehydrogenase (GAPDH), caspase-3, and cyclin-dependent kinase 5 (CDK5) [86,87]. Each S-nitrosylase targets only a set of proteins involved in particular pathways, allowing for the selectivity of their regulations [85]. There could be hundreds of more S-nitrosylases to be identified, given the total number of S-nitrosylated proteins (>3000) in cells [2,3].

In contrast, denitrosylation is a process that reverts S-nitrosylation through enzymatic reactions (Figure 1). The balance between S-nitrosylation and denitrosylation determines the overall degree of S-nitrosylation. Two denitrosylases have, so far, been well characterized: thioredoxin/thioredoxin reductase (Trx/TrxR) and glutathione/S-nitrosoglutathione reductase (GSH/GSNOR) [88,89]. Nevertheless, recent studies have identified several potential denitrosylases, including glutathione peroxidase, protein disulfide isomerase and ceruloplasmin [87]. Similar to S-nitrosylases, there could be a number of denitrosylases which are yet to be identified.

## 4. S-Nitrosylation in Diseases

### 4.1. S-Nitrosylation in Cancer

Cumulative evidence attests to the critical roles of dysregulated NO and S-nitrosylation in the pathogenesis of different types of cancer. Aberrant S-nitrosylation levels are attributed to various factors, including the altered expression of NOSs and denitrosylases, oxidative stress, hypoxia, and oncogenic mutations of target proteins [13,17,18,19,90,91,92].

Numerous s-nitrosylated proteins that contribute to the pathogenesis of different types of cancers have been identified with the help of different bioinformatic analyses (Table 1).

#### 4.1.1. S-Nitrosylation Influenced by Altered Expression of NOSs and Denitrosylases as Well as Oxidative Stress

iNOS is preferentially upregulated in many types of cancers [115,116], leading to elevated NO production and S-nitrosylation. For example, in triple-negative breast cancer cells, increased iNOS level is linked to elevated S-nitrosylation of p21^Ras^, which promotes oncogenic signaling via mitogen-activated protein kinase kinase (MEK)/extracellular signal-regulated kinase 1/2 (ERK)/ETS proto-oncogene 1 (ETS1) [117]. On the other hand, NOS functionality could, instead, be compromised in malignant cells due to the highly oxidizing environment that could degrade the essential cofactor BH_4_ to “uncouple” the NOS dimer [20,52,118,119,120,121]. In such a condition, NO production and S-nitrosylation would be downmodulated in cancer cells regardless of NOS levels. Alternatively, S-nitrosylation could be elevated in cancer cells because of downmodulation of denitrosylase. For example, the expression of a major denitrosylase GSNOR is downmodulated in ~50% of hepatocellular carcinomas (HCC). GSNOR deficiency was found to elevate S-nitrosylation and proteasomal degradation of the key DNA repair protein, O(6)-alkylguanine-DNA alkyltransferase (AGT), promoting oncogenic mutagenesis [122].

#### 4.1.2. S-Nitrosylation Influenced by Hypoxia

Hypoxia is an inducer of diverse biological events, including angiogenesis and glycolysis, which are two hallmarks of cancer. Hypoxia elevates the expression of eNOS and iNOS in endothelial cells, leading to the increase in S-nitrosylation of hypoxia-inducible factor 1 alpha (HIF1-α) at Cys520 and Cys800. S-nitrosylation of Cys520 in the oxygen-dependent degradation (ODD) domain protects HIF1-α from ubiquitin-mediated degradation and stabilizes the protein. Furthermore, S-nitrosylation of Cys800 promotes HIF1-α interaction with transcription co-factors, such as p300 and CBP, to activate the transcription of a vast array of genes, such as those involved in angiogenesis (e.g., vascular endothelial growth factor [VEGF] and TEK tyrosine kinase [TIE2, angiopoetin-1 receptor]) as well as genes involved in glycolysis (e.g., glucose transporter 1 [GLUT1] and aldolase A) [95,104,105].

#### 4.1.3. S-Nitrosylation Influenced by Oncogenic Mutations

Cancer cells and normal cells often exhibit drastically different S-nitrosylation profiles. Tan et al. performed site-specific proteomic analysis of S-nitrosylated cysteines and identified 397 unique sites in 290 unique proteins for pancreatic ductal adenocarcinoma (PDAC) vs. 25 unique sites in 25 unique proteins for adjacent normal tissues. There were only 66 shared sites in 63 shared proteins for both normal and cancerous tissues [25]. Such differential usage of S-nitrosylation sites and proteins in cancer vs. normal cells could be, in part, attributed to oncogenic mutations in malignant cells. These mutations could have destroyed the reactive cysteines, generated new reaction sites, or altered the local environment of the reaction sites to modulate the accessibility to NO and oxygen [10,15,87]. One such example is p21^Ras^ GTPase. In normal cells, p21^Ras^ is S-nitrosylated at Cys118 within the conserved NKCD motif (Asn116-Lys 117-Cys118-Asp119) involved in nucleotide binding. This S-nitrosylation facilitates guanine nucleotide exchange (from GDP to GTP) of the catalytic site for enzymatic turnover. On the other hand, in cancer cells, p21^Ras^ is often subjected to Gly12Cys and Gly13Cys mutations. This generates the two additional S-nitrosylation sites, increasing the affinity to GTP to further promote guanine nucleotide exchange and exacerbate the pro-tumor activity [123].

#### 4.1.4. Dichotomous Effects of S-Nitrosylation on Cancer

In addition to dysregulated S-nitrosylation levels in cancer, S-nitrosylation itself could elicit dichotomous effects on cancer development, further complicating matters. In some cases, S-nitrosylation inhibits tumor progression, whereas, in other cases, it promotes tumor growth [25,107,108]. Such bifurcated consequences of S-nitrosylation likely depend on the tumor types, target proteins, and specific TME.

For example, in non-small cell lung carcinoma (NSCLC) cells, S-nitrosylation of the antioxidant enzyme peroxiredoxin-2 (PRDX2) inhibits its enzymatic activity, resulting in H_2_O_2_ accumulation. This causes the activation of 5′ adenosine monophosphate-activated protein kinase (AMPK), which, in turn, phosphorylates Sirtuin 1 (SIRT1) to inhibit its deacetylase activity toward p53 and forkhead Box O1 (FOXO1) for their repression. This liberates the pro-apoptotic functions of the two tumor suppressor proteins, leading to cancer cell death [102]. If such pro-apoptotic signaling triggered by S-nitrosylation of PRDX2 takes place in neurons, however, this could lead to neurodegenerative disorders, such as Parkinson’s disease [124]. Moreover, even in the same NSCLC cell, if S-nitrosylation is targeted to ezrin, which is a linker between the plasma membrane and actin cytoskeleton, this could lead to tumor promotion. S-nitrosylation of ezrin promotes its interactions with actin and, thus, elevates cytoskeletal tension at the plasma membrane, increasing the invasiveness of cancer cells [99]. In addition, S-nitrosylation of an anti-apoptotic protein, B-cell lymphoma 2 (BCL-2), in lung cancer cells also exerts a pro-tumor function. S-nitrosylated BCL-2 binds Beclin-1, which is a critical autophagy regulator [125]. The BCL-2-Beclin-1 complex, in turn, inhibits the formation of autophagosomes and, ultimately, autophagy-mediated cell death, promoting lung cancer progression.

Likewise, in glioma cells, S-nitrosylation of ERK1/2 mitogen-activated protein kinases (MAPK) suppresses their phosphorylation and activation, promoting apoptosis [107]. In normal neurons, however, ERK1/2 plays essential roles in cell survival and should not be inhibited by S-nitrosylation [126]. Thus, S-nitrosylation instead takes place at the upstream protein p21^Ras^, which conversely activates ERK1/2 pathway for neurogenesis [127]. Moreover, even in glioma cells, if S-nitrosylation is targeted to caspase-3, this inhibits its pro-apoptotic activity and promotes tumor cell growth [128]. These examples clearly demonstrate that S-nitrosylation of different proteins in different cell types could result in contrasting effects on cell growth and survival.

Most of these studies attribute the source of the NO-group for S-nitrosylation to GSNO, which is the most abundant nitrosothiol in cells. However, in some cases, NO produced by NOS could rapidly S-nitrosylate proteins in close proximity. For example, in colorectal cancer cells, iNOS forms a complex with latent TGF-β binding protein (LTBP1) along with 6-pyruvoyl-tetrahydropterin synthase (PTPS), a critical enzyme for the biosynthesis of BH_4_—the essential cofactor of NOS. In response to hypoxia, AMPK phosphorylates PTPS, which induces PTPS to bind LTBP1, leading to the formation of the PTPS/iNOS/LTBP1 complex. Both BH_4_ and NO are produced within the same complex, which facilitates the efficient S-nitrosylation of LTBP1 to target the protein for proteasome-mediated degradation. Loss of LTBP1 then inhibits TGF-β secretion, allowing for continuous tumor cell growth under hypoxia [109]. It should be noted that TGF-β plays dichotomous roles in cancer depending on cancer type, stage, and context, which complicates the consequence of its inhibition. For example, TGF-β suppresses tumor growth in the early-stage colon, gastric, bladder, and ovarian cancer in a cell autonomous manner. In contrast, TGF-β promotes tumor growth in the advanced stage breast, esophageal, lung, and pancreatic cancer in a non-cell-autonomous manner (via interactions with the microenvironment) [129].

### 4.2. S-Nitrosylation in Other Diseases

In addition to its role in cancer pathogenesis, S-nitrosylation could also play key roles in the development of other types of diseases (Table 1). For example, as discussed above, increased levels of S-nitrosylated proteins are implicated in the progression of neurodegenerative diseases [116,119,120]. In the brain tissues of patients with Alzheimer’s disease, 45 differentially S-nitrosylated proteins have been identified [130]. In contrast, S-nitrosylation plays a beneficial role in cardio-protection. For example, upon the incidence of ischemia that lowers the oxygen supply to the cardiac muscle, a number of proteins become S-nitrosylated. This helps the heart become preconditioned for the low oxygen level and also for the upcoming reperfusion that elicits oxidative tissue injury. S-nitrosylation of these proteins not only lowers the cells’ need for oxygen and prevents their necrosis and apoptosis, but also protects the proteins from oxidation during reperfusion [131]. Moreover, in renal proximal tubules, inhibitory S-nitrosylation of enzymes involved in intermediary metabolism could protect the kidney against acute injury [132].

The above examples clearly depict dichotomous effects of S-nitrosylation on the pathogenesis of different diseases. The overall consequences of this protein modification depend on the context, cell/tissue type, target proteins, and the resulting molecular events. However, these intricate roles of S-nitrosylation remain the subject matter of further investigations. Such a complexity becomes amplified when S-nitrosylation takes place in the tissue/tumor microenvironment composed of a collection of cells and secreted proteins as we discuss in the following section.

## 5. S-Nitrosylation in the Tumor Micro-Environment

The tumor micro-environment (TME) is composed of a heterogeneous group of cells, including cancer cells, fibroblasts, vasculature, immune cells, adipocytes, and other types of cells, as well as the ECM and other secreted proteins (Figure 2) [133,134]. Crosstalk between different resident cells in the TME promotes tumor progression and contributes to the acquisition of therapeutic resistance. In particular, tumor cells could reprogram the functions of different components of TME for their own growth advantage [135,136].

In the TME, a broad variety of cell types, including immune cells such as macrophages and natural killer (NK) cells, can produce NO and exert autocrine and paracrine effects. Macrophages represent a significant source of NO in the TME. Recent studies have revealed novel roles of S-nitrosylation/denitrosylation in modifying the phenotype of tumor-associated immune cells and other types of stromal cells, such as fibroblasts and endothelial cells. Dysregulated S-nitrosylation in these cells could reshape the TME from an immuno-active to an immuno-suppressive environment. In the following section, we will discuss the effects of S-nitrosylation on different resident cells of the TME.

### 5.1. S-Nitrosylation in Tumor-Associated Immune Cells

It has been established that NO plays crucial roles in both innate and adaptive immunity. Aside from its well-known roles in anti-microbial responses, NO acts as a key regulator of tumor-associated immune cells, which can switch their immunogenicity (i.e., immuno-active vs. immuno-suppressive) in response to dynamic signal interactions in the TME. Recent studies have unveiled that such NO’s influences on immune cell functions are partly mediated through protein S-nitrosylation [38,137]. Numerous immune cell-specific S-nitrosylated proteins have been identified [138,139] (Table 2). Despite the fact that the number of these proteins continues to grow, little is known about how the S-nitrosylated proteins impact immune cell functions. The following section discusses some known examples of NO-mediated immune cell regulation that prominently affect tumor immunity.

#### 5.1.1. Tumor Associated Macrophages (TAMs)

Tumor associated macrophages (TAMs) are a heterogeneous group of macrophages that occupy more than 50% of the TME [140]. Immunogenicity of the TME is predominantly regulated by the ratio between two distinct subtypes of TAMs: M1 TAMs, which exert immuno-stimulatory responses, and M2 TAMs, which exert immuno-suppressive responses. Importantly, M1 TAMs produce high levels of NO compared to M2 TAMs [141,142]. M1 macrophages are activated by pro-inflammatory stimuli (e.g., IFN-γ, LPS) that upregulate iNOS expression to generate a burst of NO. Apart from its inherent cytotoxic effects, the anti-tumorigenic and immunogenic responses of NO produced by M1 macrophages are largely carried out by protein S-nitrosylation. Such S-nitrosylation-regulated immunological pathways include CC chemokine receptor 5 (CCR5) signaling (protein Kinase C delta [PRKCD], Receptor For Activated C Kinase 1 [RACK1] and G Protein Subunit Beta 1 [GNB1]); interleukin 12 (IL-12) production and signaling (lysozyme [LYZ], CCAAT enhancer binding protein beta [CEBP-β]); and phagocytosis (Rac Family Small GTPase 1 [RAC1], RAC2 and beta-actin [ACTB]) [138]. Despite large amounts of NO production, activated M1 macrophages are able to protect themselves against toxic levels of NO and reactive nitrogen species (RNS) by compartmentalizing nitrosative stress in the phagosomes with the help of denitrosylases, such as GSH/GSNOR and Trx/TrxR [143].

#### 5.1.2. T Cells

T cells are a major component of the adaptive immune system induced to attack target cells carrying specific antigens presented by antigen-presenting cells (APC). Tumor infiltrating T cells fall into different subtypes and exert distinct immune responses. T helper 1 (Th1) and cytotoxic (CD8^+^) T cells induce immunogenic responses, while Th2, Th17, and Treg cells induce immuno-suppressive mechanisms. Upon engagement of APC, the microtubule-organizing center (MTOC) and the associated Golgi apparatus in T helper cells translocate to the APC contact site. This induces the recruitment of the Golgi-localized eNOS to the immune synapse, a specialized intercellular site where the T cell receptor (TCR) accumulates. This activates NO production by eNOS and potentiates TCR signaling in response to the antigen binding [144]. In T cell-mediated immunity, NO signals are propagated via S-nitrosylation/denitrosylation that regulates the differentiation and activation of T cells [145]. For instance, effector T cells, including Th1 and CD8^+^ T cells are activated by NO produced from M1 TAMs. There has been increasing evidence that suggests the role of denitrosylation in T-cell activation [146]. Nitrosothiols could be denitrosylated through the conversion of GSH to GSNO, which is further reduced by GSNOR. Denitrosylation is particularly important for T cell function because the development of T cells is impaired in GSNOR-deficient mice [147]. A recent study has also shown that GSNOR-dependent denitrosylation of protein kinase B (AKT) is involved in the T cell activation during hyper-homocysteinemia (HHcy)-induced atherosclerosis [148].

In the TME, high levels of NO induce S-nitrosylation of the chemokine CCL2, which prevents the recruitment of CD8^+^ T cells to tumor tissues and renders the TME immuno-suppressive [157]. In addition, upregulation of iNOS along with arginase 1 (ARG1) in myeloid-derived suppressor cells (MDSCs) leads to arginine depletion in the TME, inducing S-nitrosylation of the T cell receptor (TCR) and T-cell apoptosis [156].

#### 5.1.3. Natural Killer (NK) Cells

NK cells are granular lymphocytes of the innate immune system that induce cytotoxicity against virally infected or tumorigenic cells. Activation of NK cells is a complex process, which requires stimulation by various cytokines (i.e., IL-2, IL-12, and IL-15) and a shift of the signaling cascade from inhibitory receptors (human leukocyte antigen [HLA]-A, -B, CD48) to activating receptors (NKp46, NKG2D, and NKp30) [182]. Upon activation, NK cells produce high levels of NO that facilitates their cytolytic function [183,184]. Furthermore, activated NK cells regulate the activation and maturation of T cells, dendritic cells, and macrophages, which subsequently leads to an improved anti-tumor response [185]. Apart from being an effector molecule, NO also regulates the activation and survival of NK cells. Nevertheless, the exact impact of NO on NK cells and the underlying mechanisms remain to be determined [183]. The impact of S-nitrosylation on some of the crucial immune regulatory proteins is discussed below.

#### 5.1.4. Nuclear Factor Kappa B (NF-κB)

NF-κB is an evolutionarily conserved family of transcription factors that form a heterodimer to regulate inflammatory and immune responses, cell proliferation, and apoptosis. Therefore, the NF-κB signaling pathway is considered to be a crucial target for cancer therapeutics [14]. Different cell surface receptors such as the tumor necrosis factor (TNF) receptor, Toll-like receptor (TLR), and T cell/B cell receptors activate NF-κB signaling in response to diverse external stimuli. In the cytoplasm, a NF-κB heterodimer is in a complex with the inhibitor of NF-κB (IκB) and remains inactive. In response to an activation signal, IκB becomes phosphorylated and is targeted for proteasomal degradation, liberating NF-κB which, then, translocates to the nucleus [186,187]. In the nucleus, NF-κB transactivates a multitude of target genes including cytokines (interferon gamma [IFNγ], IL1α and IL12); immunoreceptors (CD48, major histocompatibility complex I [MHC I] and MHCII); apoptotic regulators (Fas, FasL and BCL-xL); and growth factors (granulocyte-colony-stimulating factor [G-CSF] and macrophage-CSF [M-CSF]) that are essential for regulating immune responses and tumor growth [188].

S-nitrosylation has been shown to regulate NF-κB activity by targeting multiple components [158]. In macrophages and T cells, S-nitrosylation of IκB kinase (IKKβ) at Cys179 residue represses its activity to phosphorylate IκB for proteasomal degradation (Figure 3A) [14,171,172]. As a result, the NF-κB-IκB complex remains inactive in the cytoplasm. Furthermore, p50 and p65 subunits of the NF-κB heterodimer are also inactivated via S-nitrosylation at Cys62 and Cys38, respectively (Figure 3A) [158,189]. Inactivation of these subunits inhibits their DNA binding for gene transcription. In contrast, S-nitrosylation has also been reported to activate the downstream molecules of NF-κB signaling, such as p21^Ras^, nicotinamide adenine dinucleotide phosphate (NADPH) oxidase and thioredoxin [158]. Such conflicting results could be attributed to dichotomous effects of NO and S-nitrosylation in a cell type-dependent and concentration-dependent manner.

#### 5.1.5. Signal Transducer and Activator of Transcription 3 (STAT3)

STAT3 is a transcription factor which acts at the intersection of many signaling pathways induced by different cytokines (e.g., IL-6, IL-11, IFNα) and growth factors (human epidermal growth factor receptor 2 [HER2], epidermal growth factor [EGFR] and VEGF) [190,191]. STAT3 is constitutively active in many types of cancer with high frequency [192]. While being highly activated in both tumor-associated immune cells and tumor cells, STAT3 plays crucial roles in downregulating anti-tumor responses to promote tumorigenesis [193,194]. In immune cells, STAT3 elevates the expression of anti-inflammatory factors (IL-10, transforming growth factor beta [TGFβ] and cyclooxygenase 2 [COX2]) and inhibits the expression of pro-inflammatory factors (tumor necrosis factor alpha [TNFα] and IFNγ) (32). Thus, STAT3 activation leads to the downmodulation of dendritic cell maturation, M1 TAM polarization, and Th1-type immune responses, while promoting the expansion of M2 TAMs, Th2 cells, Tregs and MDSCs [194,195]. In addition, STAT3 activation in tumor cells elevates the expression of genes linked to proliferation (cellular myelocytomatosis oncogene [c-MYC], Cyclin D1, BCL-xl) and angiogenesis (VEGF, hepatocyte growth factor [HGF]); and metastasis (matrix metalloproteinase-2 [MMP2], MMP9, and twist-related protein 1 [TWIST1]), promoting malignant progression [190,191]. Therefore, STAT3 is a newly emerging target of immunotherapies for many types of cancer [192,196]. Recent studies unveiled that STAT3 activity could be inhibited by S-nitrosylation at Cys259 in tumor-associated immune cells (Figure 3B) [32,181,182]. This finding suggests that S-nitrosylation-mediated suppression of STAT3 could improve the immunogenicity of the TME.

#### 5.1.6. Caspases

Caspases are a family of proteases consisting of 11 members that cleave target proteins at the C-terminus of aspartate residues. These proteolytic enzymes are categorized into three functional groups, including inflammatory caspases (Caspase-1, Caspase-4 and Caspase-5); initiator caspases (Caspase-8, Caspase-9 and Caspase-10); and executioner caspases (Caspase-3, Caspase-6 and Caspase-7) [197]. Apart from their well-known function in regulating programmed cell death and inflammation, caspases are also known to regulate cell proliferation, differentiation and aging [198,199]. More importantly, they are implicated in many hallmarks of cancer, such as evasion of cell death, sustained proliferation, immune evasion and tumor-promoting inflammation, making caspases an ideal target for anti-cancer therapies [200]. NO was found to regulate the activity of many of these caspases primarily through S-nitrosylation/denitrosylation of the cysteine groups in their active sites [14]. The impact of S-nitrosylation on two important caspases that regulate the TME are discussed below.

Caspase-1 is an inflammatory caspase and is predominantly involved in the differentiation, activation and polarization of phagocytic cells, such as macrophages [198,201]. This enzyme is activated within inflammasomes, which are the cytosolic complexes of multi-protein immune receptors (i.e., pattern recognition receptors). Caspase-1 mediates immune responses by promoting the maturation and secretion of pro-inflammatory cytokines, such as IL-1β and IL-18, and by regulating NF-κB signaling [32,198]. S-nitrosylation of caspase-1 at the catalytic site cysteine (Cys285) inhibits its activation, lowering the production of IL-1β and IL-18 by inflammasomes (Figure 4A). In particular, S-nitrosylation of caspase-1 inhibits the functions of NLR family pyrin domain containing 3 [NLRP3] inflammasome, which is an activation platform of caspase-1 [14,163]. It was shown that inhibiting NLRP3 inflammasome through caspase-1 S-nitrosylation was able to suppress angiogenesis, invasion and metastasis of melanoma and breast cancer cells [186,189,190].

Caspase-3 is an executioner caspase with a well-known function in regulating apoptosis. Located within the cytoplasm and the mitochondrial intermembrane space in mammalian cells, caspase-3 becomes activated by extrinsic death signals transduced via death receptors, such as TRAIL and Fas [14,160]. Activation of this executioner caspase requires cleavage by initiator caspases, such as caspase-8 and caspase-9. Upon activation, caspase-3 undergoes a conformational change. The active enzyme then targets key structural and regulatory proteins associated with cell survival (e.g., poly [adenosine diphosphate (ADP-ribose)] polymerase [PARP], EGFR and gelsolin), leading cells to apoptosis [202,203]. However, S-nitrosylation of caspase-3 at the catalytic site cysteine (Cys163) inhibits its activity (Figure 4B) [14]. Such anti-apoptotic effects of caspase-3 S-nitrosylation are utilized by tumor-associated immune cells to improve their anti-tumor responses. On the other hand, suppression of apoptosis in tumor cells is a key driving force for tumorigenesis [204]. Nevertheless, recent studies revealed that activation of caspase-3-mediated apoptosis in cancer cells could also induce the “Phoenix Rising” pathway that produces biochemical signals to regenerate tumor tissues. As such, elevated caspase-3 levels in breast cancer are associated with a worse clinical outcome [205,206]. Therefore, caspase-3 inhibition in certain conditions could have anti-tumor effects rather than pro-tumor effects.

### 5.2. S-Nitrosylation in Endothelial Cells

The first known biological role of NO was the regulation of endothelial cell functions, such as vascular tone, migration and permeability [207]. In endothelial cells, NO is produced by eNOS, which mediates S-nitrosylation of multiple proteins regulating protein trafficking, cell migration, redox state and cell cycle. Decreased levels of S-nitrosylated proteins in endothelial cells have been linked to a variety of disease conditions, including congestive heart failure and hypertension. Conversely, increased levels of S-nitrosylated proteins in endothelial cells could promote tumor pathogenesis by triggering angiogenesis and tumor cell attachment to endothelium, allowing for tumor growth and metastasis [208,209]. Below are some examples of endothelial proteins that act to promote tumor progression upon S-nitrosylation.

Vascular endothelium-cadherin (VE-cadherin), expressed specifically in endothelial cells, is located at adherens junctions between endothelial cells. It associates with several cytoplasmic proteins, including α-catenin, β-catenin, γ-catenin and δ-catenin (p120), to maintain the endothelial barrier. In glioblastoma, however, IL-8 secreted by tumor cells triggers eNOS-mediated NO production and then S-nitrosylation of VE-cadherin and p120, which impairs their associations and induces hyperpermeability of blood vessels [149].

VEGF plays a major role in inducing vascular permeability with the help of eNOS. VEGF promotes eNOS-dependent S-nitrosylation of β-catenin at Cys619, leading to its dissociation from VE cadherin. This, in turn, triggers the disassembly of the adherens junction complex, elevating the vascular permeability [151]. In fact, eNOS-deficient mice were found to be defective in vascular permeabilization even in the presence of VEGF, suggesting the critical role of eNOS in executing the VEGF signal [151].

HIF-1α is a subunit of a heterodimeric transcription factor HIF1, which induces angiogenesis in response to hypoxia to maintain tissue metabolism [210,211,212]. Hyperactivation of HIF-1α plays an important role in cancer metastasis [90,213,214,215,216]. For example, S-nitrosylation of HIF-1α at Cys533 elevates the activity, resulting in the increased expression of various angiogenic factors that stimulate angiogenesis and promote cancer metastasis [152].

Given the adversary effects of excessive S-nitrosylation of endothelial proteins, eNOS inhibitors have been tested for their efficacies of minimizing tumor metastasis. A study by Gao and colleagues showed that knocking-down eNOS expression or treatment with eNOS inhibitors, 1400 W and L-NIO, suppressed angiogenesis and compromised colorectal cancer progression [217]. The same group also showed that Celastrol, a phytochemical which inhibits NOS activity, impaired angiogenesis in colorectal cancer [218].

### 5.3. S-Nitrosylation in the Extracellular Matrix (ECM)

ECM is the acellular component of the TME, which provides not only the structural support, but also a variety of signals to regulate tissue homeostasis [219]. The ECM constituents, also known as the “core matrisome”, are about 300 proteins including fibrous proteins, growth factors, ECM-modifying enzymes and other ECM-associated proteins [220].The ECM components dynamically remodel in response to varying environmental cues. The ECM then regulates a variety of cellular functions, including cell survival, differentiation, maintenance of tissue architecture and migration [220].

Normal epithelial cells require attachments to the ECM for their growth, differentiation and survival. If epithelial cells detach from the ECM, they undergo programmed cell death through a phenomenon termed anoikis [221]. During cancer progression, however, malignant cells acquire the ability to survive and metastasize without the need of attachment to the ECM—anoikis-resistance [221,222,223]. Such anoikis-resistance of cancer cells is partly attributed to S-nitrosylation of various proteins and is, otherwise, associated with the ECM, as discussed below [131,217,218].

Integrin αβ-heterodimers are essential cell surface receptors that not only mediate cell adhesion to the ECM, but also are involved in transduction of various biochemical and mechanical signals [224,225,226]. During the formation of cell-ECM adhesions, integrin-ECM interactions trigger their clustering and induction of Src kinase to phosphorylate focal adhesion kinase 1 (FAK1). This causes the assembly of focal adhesion complexes and their linkages to cytoskeletal networks [227]. These adhesion complexes, in fact, play essential roles in the induction of anoikis. However, in prostate cancer cells with iNOS overexpression, the α6-integrin subunit becomes S-nitrosylated at Cys86, which causes a shift of its dimerization partner from the canonical β4-integrin to β1-integrin. This lowers the number of integrin α6β4 heterodimers (laminin receptors) on the cell surface and the cells’ ability to bind the ECM, inducing cell migration and anoikis-resistance [106]. In addition to integrins, Src kinase could also be S-nitrosylated at Cys498, which induces autophosphorylation of the protein and promotes cells’ invasiveness and anoikis-resistance [93,228]. This mode of anoikis-resistance plays a role in estrogen-driven tumor progression.

Caveolin-1 (CAV-1) is a major structural component of caveolae, a subset of lipid rafts in the plasma membrane involved in endocytosis, ECM organization, mechano-sensing, and biochemical signaling. CAV-1 physically associates with eNOS for mutual suppression. CAV-1 binding keeps eNOS in the inactive state [229,230]. Once eNOS becomes activated in response to a stress, such as a mechanical stress, NO S-nitrosylates CAV-1 at Cys156 to target the protein for proteasomal degradation [231,232]. CAV-1 is elevated in several metastatic cancers. However, cancer-associated caveolae are often dysfunctional [233,234]. It is yet to be determined whether this is, in part, attributed to aberrant S-nitrosylation of CAV-1.

Transglutaminase 2 (TG2) is a multi-functional enzyme present in the cytosol and ECM. TG2 is highly expressed in stromal cells, such as endothelial cells, fibroblasts and monocytes/macrophages. Cytosolic TG2 primarily acts as a GTPase, while extracellular TG2 catalyzes deamidation and cross-linking of ECM proteins to regulate the tensile properties of tissues [235]. TG2 becomes S-nitrosylated to suppress the activity when the NO levels are elevated [235]. However, in aged vasculatures where the NO levels are low, TG2 S-nitrosylation is compromised, leading to excessive crosslinking of matrix proteins and tissue stiffening [236]. Importantly, TG2 is highly elevated in various types of cancer and plays major roles in establishing stiff ECM that exacerbates tumor progression. Thus, TG2 is an emerging therapeutic and diagnostic target for cancer and could be inhibited by S-nitrosylation [237].

### 5.4. S-Nitrosylation in Tumor Microbiome

The TME, as described earlier, consists of innate and adaptive immune cells, which is a network of blood and lymphatic vessels and other types of stromal cells. Furthermore, recent findings unveiled that the microbiome serves as an additional core component of the TME and impacts tumor progression [238,239,240,241]. A comprehensive analysis of the tumor microbiome by Nejmen et al. found that metabolically active bacteria live intracellularly in both cancer and immune cells and could affect the TME. They reported that the microbial composition varied according to tumor type and tissue origin (lung, breast, ovary, pancreatic, melanoma, brain and bone). In addition, there were close similarities of metabolic profiles between bacteria and host tumor cells, suggesting that these bacteria play critical roles in the tumor phenotype [238]. Microbiome compositions could also affect the TME by modulating the host immune response. Riquelme et al. transplanted human fecal microbes from PDAC patients into mice by oral gavage, and later xenografted these mice with cancer cells. They found that PDAC-derived microbes modulated the host immune system and exacerbated tumor development [239].

Better understanding of microbial biology and its influence on the TME would significantly impact the development of new cancer treatments. For example, a recent study reported that modified bacterial strains engineered to migrate to tumor hypoxic sites could be utilized to deliver anti-cancer agents to necrotic sites of tumors to overcome the difficulty in drug delivery owing to aberrant tumor vasculature [242]. Recently, Seth et al., using *C. elegans* as in vivo models, unraveled that NO-producing bacteria (i.e., *Bacillus. Subtilis*) in the gut microbiome largely influence the host physiology by S-nitrosylating host proteins [243]. This is the first-time demonstration that NO, by means of S-nitrosylation, could serve as a common language for interspecies communications between gut bacteria and host cells. A total of 924 host proteins were found to be S-nitrosylated by bacterial NO. About 200 of them were involved in metabolism while other proteins were involved in the maintenance of immunity, suggesting the role of bacterial NO in modulating the host metabolism and immunity [243]. It is, however, yet to be examined whether such NO-mediated interspecies communications take place in mammalian guts, where the host cells are separated from the gut microbes by the mucosal barrier.

Furthermore, recent studies have also unveiled the essential functions of the microbiota of various tissues/organs other than the gut. For example, the microbiota of the breast is composed of seven phyla, including *Firmicutes* (e.g., *Bacillus* spp.) and *Actinobacteria* (e.g., *Adlercreutzia* spp.), which produce NO and NOS cofactor BH_4_, respectively [244]. These bacteria are also enriched in breast milk, indicating their roles not only in breast functions, but also in neonatal development [245]. Furthermore, breast milk is the major source of maternally transferred microbes, accounting for about 40% of the gut bacteria of newborns in the first month of life [246]. However, it is yet to be determined whether these microbial contributions are attributed to their NO production. As expected, the microbiota of the breast is dramatically altered in both benign and malignant tissues compared to the normal breast [247]. As such, the discovery of microbial NO that could S-nitrosylate host proteins has added another factor of complexity in the regulation of the TME. Too much or too little NO production by the microbiome may dysregulate host protein functions and contribute to cancer pathogenesis (Figure 5). Such information could be utilized to develop novel strategies for cancer treatment.

## 6. S-Nitrosylation in Anti-Cancer Therapy

It is now evident that aberrant S-nitrosylation plays a key role in cancer development [248,249,250,251,252]. The altered S-nitrosylation levels are likely attributed to the dysregulated expression or function of NOSs (specifically iNOS) and denitrosylases (such as Trx and GSNOR) as well as oxidative stress, hypoxia and oncogenic mutations of target proteins. However, given that most S-nitrosylation inactivates the target, the biological consequence of this modification largely depends on the primary function of the protein. For example, S-nitrosylation of proteins with tumor-suppressive functions (e.g., caspases and PTEN) could exacerbate tumor development. In contrast, S-nitrosylation of proteins with tumor-promoting functions (e.g., NF-ĸB and AKT) could suppress tumor progression [209,253,254,255]. Such dichotomous effects of S-nitrosylation become apparent when different cancer types are compared. Accordingly, a therapeutic strategy to either reduce or increase S-nitrosylation could be decided based on the type and nature of cancer as discussed below [21] (Table 3).

### 6.1. Reducing S-Nitrosylation

Pharmacological inhibition of NOS is the most commonly used approach for reducing S-nitrosylation. Especially, 1400W (iNOS inhibitor), L-NAME and L-NMMA (pan NOS inhibitors) have shown preclinical feasibility for the treatment of triple-negative breast cancer (TNBC), resulting in the reduction of cell proliferation and cell motility [256]. In particular, the combinatory use of a NOS inhibitor with a chemotherapeutic agent, docetaxel, was found to improve the cytotoxicity of the drug in docetaxel-resistant TNBC cells by activating activation of apoptosis signal regulating kinase (ASK1) [257]. iNOS inhibitors have also been utilized for the treatment of liver tumors that have increased iNOS expression and reduced expression of a denitrosylase, GSNOR. These tumors were found to have hyper-S-nitrosylation of AGT, a DNA damage repair protein, which targets the protein for degradation. In these tumors, the iNOS inhibitor was able to rescue the AGT activity and block mutagenesis [258]. Another example is MDA-7/IL-24, which is a tumor suppressive cytokine currently in the early stages of FDA pre-IND drug trials [259]. This cytokine induces apoptosis of various types of tumors through denitrosylation of an anti-apoptotic protein BCL-2, targeting the protein for ubiquitination and degradation. (S-nitrosylation of BCL-2 at Cys158 and Cys229 was found to be a major mechanism to suppress apoptosis of tumor cells under stress.) This pro-apoptotic activity of MDA-7/IL-24 was shown to be mediated through both the decrease of iNOS expression and increase of a denitrosylase TRXR1 [171,260].

### 6.2. Increasing S-Nitrosylation

NO donors and NOS-inducing drugs, such as SNP, GSNO, NO-ASA (NO-releasing Aspirin) and JSK (NO pro-drug), are utilized to increase S-nitrosylation levels for cancer treatment [30,261,265,266]. This approach is based on the well-established notion that increased nitrosative stress could trigger growth inhibition and cytotoxicity in tumor cells. One of the key mechanisms by which NO donors exert such anti-tumor effects is elevated S-nitrosylation. For example, S-nitrosylation of ERK1/2 at Cys183 impaired its phosphorylation and mitogenic activity, leading to the growth inhibition of glioma cells and apoptosis of breast cancer cells [107]. Moreover, GSNO treatment resulted in increased STAT3 S-nitrosylation and inhibited the growth of ovarian cancer [105] as well as head and neck squamous cell carcinoma [261]. In addition, an NO donor, glyceryl trinitrate (GTN), was shown to induce S-nitrosylation of inhibitor of apoptosis (cIAP) at Cys571 and Cys574, which led to the assembly of a death complex in colon and breast cancer cells [262]. Among these NO donors, NO-nonsteroidal anti-inflammatory drugs (NO-NSAIDs), including NO-ASA, NONO-ASA and NO-naproxen, are the most widely used for cancer treatment [266]. These NO-releasing compounds are utilized by themselves or in combination with other chemotherapeutic agents to induce apoptosis of drug-resistant cancer cells [267]. The anti-cancer effects of NO-NSAID involve the S-nitrosylation and subsequent inactivation of various pro-tumor proteins, including NF-κB and β-catenin [265].

Dysregulated S-nitrosylation levels have been linked to metabolic reprogramming, in part owing to altered mitochondrial functions, in cancer cells and stroma cells [268,269]. In mitochondria, six out of a total of eight enzymes in the tricarboxylic acid (TCA) cycle are targeted for S-nitrosylation. These six enzymes are aconitase, citrate, succinyl-CoA synthase, isocitrate, α-ketoglutarate dehydrogenase and succinate dehydrogenase [270,271,272]. S-nitrosylation of these TCA enzymes lowers the production of metabolic intermediates and energy necessary for most cellular activities. In addition, S-nitrosylation of the complex I (subunit ND3), IV, and V (ATP synthase) in the mitochondrial electron transport chain (ETC) lowers the electron fluxes and respiratory capacity of mitochondria [273,274,275]. In contrast, succinate dehydrogenase (SDH, i.e., complex II) is not targeted for S-nitrosylation. SDH could instead be subjected to cancer-associated mutations and be utilized as a target for cancer therapy. It was found that the efficacy of an SDH targeting drug (mitocans) for liver cancer treatment could be improved by increasing S-nitrosylation of mitochondrial chaperone TNF receptor associated protein 1 (TRAP1) at Cys50 by inhibiting a denitrosylase GSNOR. TRAP1 is highly expressed in different types of cancer and regulates metabolic rewiring [180,276,277]. S-nitrosylation of TRAP1 causes its degradation, destabilizing SDH and inducing apoptosis of cancer cells [181,278,279].

### 6.3. Challenges in S-Nitrosylation-Based Anti-Cancer Therapy

Despite the emerging cancer therapeutics based on S-nitrosylation, the efficacy of this approach is challenged by the multifaceted roles of S-nitrosylation in cancer. As discussed above, S-nitrosylation could exert either pro-tumor or anti-tumor effects depending on different parameters, including the context, tumor type, target cell, and target protein. Since pharmaceutical agents that modulate S-nitrosylation levels could affect all different cells in the TME, as well as in the body, they could produce adverse off-target effects that offset the advantage. For example, many types of NO-donors are utilized in S-nitrosylation-based anti-cancer treatment. These NO donors could be conjugated to different drugs, such as NSAIDs and doxorubicin, to improve the cytotoxicity [280,281,282]. However, these NO donors rapidly release NO by simply reacting with water. Such uncontrolled NO release would affect not only tumor cells but also normal cells throughout the body. Thus, the development of a tumor cell-targeted delivery system for these drugs is essential for moving this field forward [281]. In fact, there have been several studies demonstrating the efficacy of utilizing nanoparticles and liposomes to specifically deliver NO donors to tumors [281]. There have been, in fact, concerted efforts to improve the precision targeting of these cancer drugs that modulate S-nitrosylation levels. For example, in recent decades, isoform-specific NOS inhibitors have been engineered to replace the previously developed pan-NOS inhibitors, aiming to mitigate the side effects. Additional promising approaches would be advances in biomaterials and the combination of S-nitrosylation-based cancer drugs with radiotherapy, immunotherapy, and chemotherapy.

## 7. Conclusions

Endogenous NO is unstable and has a rather short half-life [39]. NO, which is synthesized in different tissues, may diffuse across cell membranes, and exert its biological function. This NO combines with the heme group of soluble guanylate cyclase (sGC)—the first intracellular NO receptor—and activates sGC to produce cyclic guanosine monophosphate (cGMP), which is a unique second messenger molecule in cells [283]. However, increasing evidence indicates that NO performs a variety of biological functions through cGMP-independent S-nitrosylation of proteins. Various physiological functions are determined by the degree of S-nitrosylation in different tissues.

In the past two decades, S-nitrosylation has garnered considerable attention for its multifaceted roles in regulating diverse signaling events in the TME to regulate cancer development [248,249,250,251,252]. A recent study reported nitrosoproteome in PDAC patient samples and revealed that many of these S-nitrosylated proteins are involved in the regulation of the cell cycle, focal adhesions, adherent junctions, and cytoskeletal functions [24]. As we described in this review, S-nitrosylation modulates the activities of different resident cells in the tissue/TME, including immune cells (macrophages, T cells, and NK cells) and other types of stromal cells (endothelial cells). It is well accepted that the levels of S-nitrosylation are aberrant in most components of the TME. However, strategies to normalize S-nitrosylation levels for cancer treatment depend on cancer types/origins and cancer-causing proteins, which are either hyper-S-nitrosylated or hypo-S-nitrosylated. Considering such a complexity, the development of new precision cancer medicine could be aimed at restoring the physiological S-nitrosylation level of a particular protein for each cell type of the TME.

## Figures and Tables

**Figure 1 ijms-22-04600-f001:**
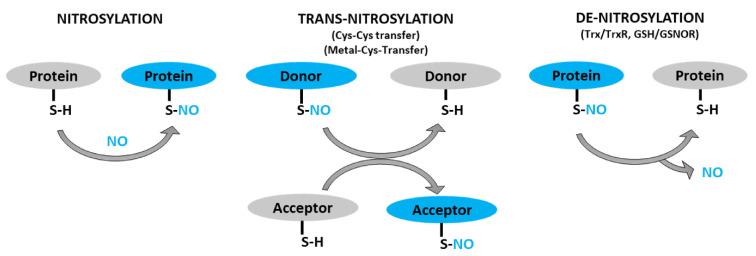
Schematic representation of protein S-nitrosylation, transnitrosylation, and denitrosylation. *S-nitrosylation*: covalent addition of NO group to the thiol (SH) group of a cysteine residue to form S-nitrosothiol (SNO). *Trans-nitrosylation*: transfer of NO moiety from donor to acceptor protein. *Denitrosylation*: removal of NO group from an already S-nitrosylated protein by denitrosylase (Trx/TrxR, GSNO/GSNOR).

**Figure 2 ijms-22-04600-f002:**
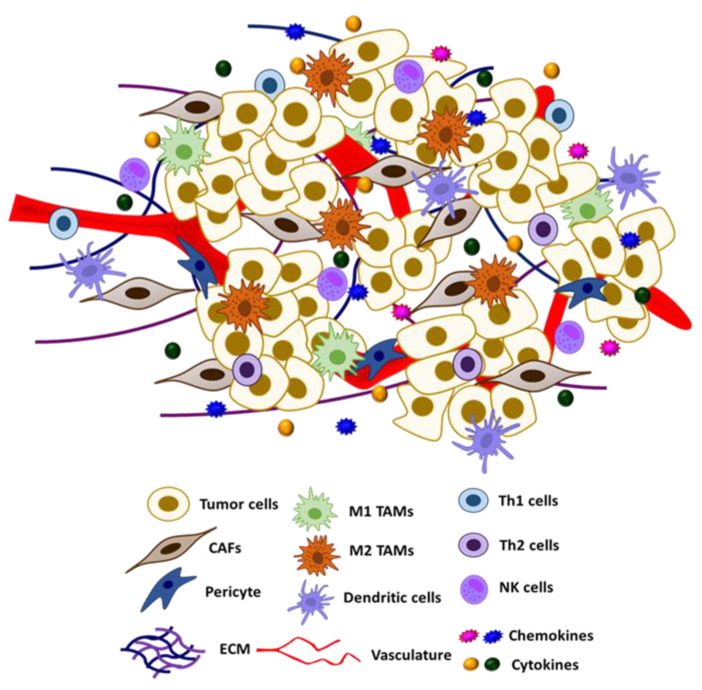
Schematic representation of tumor microenvironment and its constituents.

**Figure 3 ijms-22-04600-f003:**
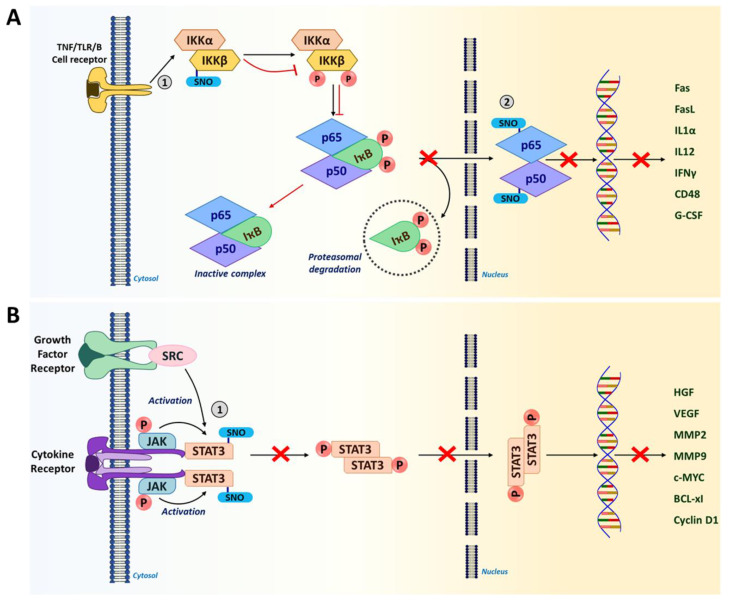
S-nitrosylation in NF-κB and STAT3 signaling pathways. (**A**) IKKβ and NF-κB subunits (p65 and p50) are S-nitrosylated in the NF-κB signaling pathway. S-nitrosylation of IKKβ at Cys179 prevents the phosphorylation of IκB and subsequent proteasomal degradation. This results in the inactive NF-κB-IκB complex sequestered in the cytosol. Furthermore, S-nitrosylation of NF-κB subunits p65 (Cys38) and p50 (Cys62) inhibits their DNA binding, in turn, preventing the transcription of NF-κB target genes including a number of pro-inflammatory cytokines. (Red arrows show the impact of S-nitrosylation.) (**B**) In tumor cells, when STAT3 is phosphorylated, it leads to the expression of genes related to proliferation and angiogenesis that promote tumor progression. However, S-nitrosylation of STAT3 at Cys259 leads to its inactivation by preventing phosphorylation. This could lead to improved immunogenicity in the TME. (1 and 2 represent the site of S-nitrosylation in the pathway).

**Figure 4 ijms-22-04600-f004:**
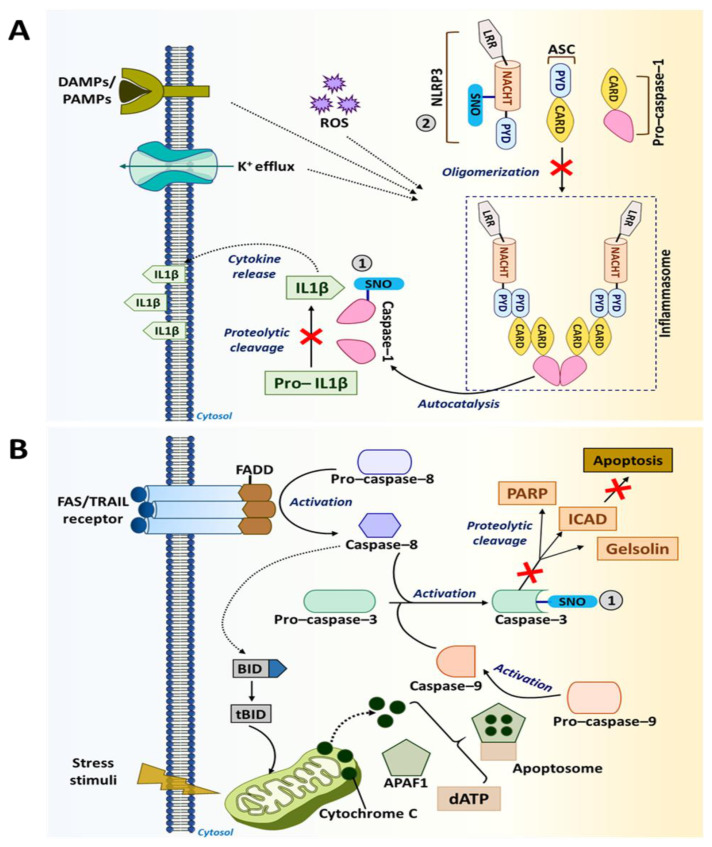
S-nitrosylation in caspase pathways. (**A**) Caspase-1 promotes the maturation and secretion of pro-inflammatory cytokines, such as IL-1β mediating immune response. However, S-nitrosylation of caspase-1 at Cys285 inhibits the maturation and secretion of IL-1β. NLRP3 inflammasome helps in the activation of caspase-1. Additionally, NLRP3 also undergoes S-nitrosylation and becomes inactive. (**B**) Cell death receptors, such as Fas/TNF-related apoptosis-inducing ligand [TRAIL], activate caspase-3-mediated apoptotic signaling in the presence of caspase-8 and caspase-9. However, S-nitrosylation of caspase-3 at the catalytic site cysteine (Cys163) causes the inhibition of its apoptotic activity.

**Figure 5 ijms-22-04600-f005:**
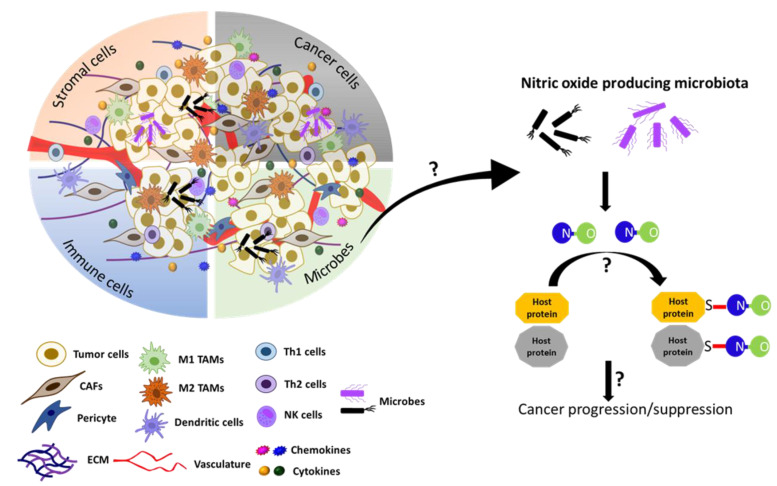
Model speculating the role of microbial NO in tumor progression or suppression in the tumor microenvironment. Recently, the gut microbiota in *C. elegans.* has been shown to produce NO and regulate the S-nitrosylation of host proteins and affect the host development [243]. The TME also contains microbes in addition to cancer cells, stromal cells, immune cells, and acellular components [238,239,240,241]. We speculate here that, if the microbiome is present in the TME, it could produce NO and regulate the tumor progression or suppression by affecting S-nitrosylation of cellular proteins in the TME.

**Table 1 ijms-22-04600-t001:** S-nitrosylated proteins linked to different diseases.

Protein	Associated Disease	Status of S-Nitrosylation in Disease	Reference
**Cancer**			
C-Src	Breast cancer	Increased	[93]
H-Ras		Increased	[94]
COX2		Increased	[95]
HIF1α	Breast cancer	Decreased	[90,96]
Galectin-1	Lung cancer	Increased	[97,98]
Ezrin	Lung cancer	Increased	[99]
BCL-2		Increased	[100]
Caveolin-1		Increased	[101]
Peroxiredoxin-2		Decreased	[102]
Rac1	Pancreatic cancer	Increased	[25]
Rac2		Increased	
STAT1		-	
PGK1		-	
RB		-	
PFKM	Ovarian cancer	Increased	[103]
Caspase-3		Decreased	[104]
STAT3	Ovarian cancer Pancreatic cancer Head and neck cancer	Increased	[25,105]
Androgen receptor	Prostate cancer	Increased	[28]
Integrin α6		Increased	[106]
ERK1/2	Glioma	Decreased	[107]
Keap1	Colon cancer	Increased	[108]
LTBP1	Colorectal cancer	Increased	[109]
**Neurodegenerative Disease**			
PTEN	Alzheimer’s disease	Increased	[110]
CDK5		Increased	
APOE		Increased	
DNM1L		Increased	
Tubulin		Increased	
SOD2		-	
MMP9	Cerebral ischemia	Increased	[111]
NMDA Receptor	Dementia	Increased	[112]
**Cardiovascular Disease**			
NSF	Pulmonary arterial hypertension	Decreased	[113]
NOS3		Decreased	
CLTC		Decreased	
Thioredoxin 1 (Trx)	Myocardial ischemia	Increased	[114]

**Table 2 ijms-22-04600-t002:** Different S-nitrosylated proteins in the resident cells of the TME and their impact on cancer.

Protein	Signaling Pathway	Impact on Protein	Physiological Impact of S–Nitrosylation during Cancer	S-Nitrosylation Site (* Potential Site)	Ref
**Endothelial cells**					
VE–Cadherin	Disassembled adherens junction between endothelial cells	Induced phosphorylation and internalization	Increased cell migration; hyperpermeability	–	[149]
p120	Disassembled adherens junction between endothelial cells	Inhibited binding with β–Catenin	Increased cell migration	Cys579	[150]
β–Catenin	Disassembled adherens junction between endothelial cells	Inhibited binding with p120	Increased cell migration	Csy619	[151]
HIF1–α	Activated HIF1 signaling pathway	Increased activation and stability	Increased angiogenesis and cancer metastasis	Cys533	[152]
Dynamin	Promoted clathrin–dependent endocytosis of β–Adrenergic receptor	Increased self–assembly and GTPase activity	Increased angiogenesis	Cys86, Cys607	[153,154]
MKP7	Activated JNK3 signaling pathway	Inhibited phosphatase activity	Increased angiogenesis and migration	Cys244 *	[13,155]
**Immune cells**					
T cell receptor	–	–	Decreased T cell proliferation and migration; increased T cell apoptosis	–	[156]
CCL–2	Reduced activity of CCR2/CCL2 signaling pathway	Decreased protein expression.	Decreased T cell infiltration	–	[157]
NF–kB	Inactivated NF–kB signaling pathway	Inhibited DNA binding activity	Decreased inflammation	Cys179	[158]
STAT3	Inactivated STAT3 signaling pathway	Inhibited activation	Decreased immune inflammatory response	Cys259	[159]
Caspase–1	Inhibited activation of NLRP3–Caspase–1 inflammasome	Inhibited activation	Decreased immune inflammatory response	Cys285	[32]
Caspase–3	Inhibited downstream activation of Caspase–3 signaling	Inhibited activation	Decreased cancer cell apoptosis	Cys163	[160]
JNK1	Inhibited activation of JNK signaling pathway	Inhibited activation	Decreased inflammation	–	[32]
NLRP3	Inhibited activation of NLRP3–Caspase–1 inflammasome	Inhibited activity	Decreased immune inflammatory response	–	[161,162,163]
NOS2	–	Suppressed activity	Decreased immune inflammatory response	–	[32]
ARG1	–	Increased protein stability	Increased immunosuppressive response	Cys303	[164]
**Others (e.g., tumor cells)**					
p21^Ras^	Promoted Guanine Nucleotide Exchange and activate downstream signaling pathways	Promoted protein activity	Increased Ras induced tumor growth	Cys118	[94,165]
p21^Ras^ (oncogenic)	–	–	Increased tumorigenic growth	Gly12Cy, Gly13Cys	[123]
COX2	–	Stimulated protein activity	Increased inflammation	Cys526	[95,166]
EGFR	Inhibited activation of EGF/EGFR signaling pathway	Impaired tyrosine kinase activity	Decreased tumorigenic growth	–	[167]
OGG1	Reduced activity of BER (Base excision repair) pathway	Inhibited activity	Impaired DNA damage repair response	–	[168]
AGT1	Suppressed activity of direct DNA repair pathway	Promoted protein degradation	Impaired DNA damage repair response	Cys145	[169]
Apo2L/TRAIL receptor DR4	Inhibited activation of death receptor signaling pathway	Inhibited activity	Decreased cancer cell apoptosis	Cys336	[170]
Bcl–2	–	Promoted protein stability	Decreased cancer cell apoptosis	Cys158, Cys229	[100,171,172]
ERK	Suppressed activity of ERK/MAPK pathway	Suppressed kinase activity	Increased cancer cell apoptosis	Cys183	[173]
HDAC2	Induced protein release from chromatin.	Increased acetylation activity.	Increased histone acetylation	Cys262, Cys274	[174,175]
PTEN	Activated downstream Akt signaling pathway	Inhibited enzymatic activity	Increased tumor progression	–	[176]
Src	Activated oncogenic signaling pathways (Akt, c–MYC)	Increased kinase activity	Increased tumor growth and proliferation	Cys498	[93]
Androgen receptor	Suppression of androgen receptor signaling	Suppressed DNA binding activity	Increased tumor growth	Cys601	[28]
Integrin α6	–	Suppressed binding to ECM	Increased cell migration	Cys86	[106]
Caveolin–1	–	Prevented proteasomal degradation	Increased tumor progression	Cys156	[33]
p53	–	Induced activation	Increased transactivation of antioxidant genes	–	[177]
MDM2	–	Inhibited activity	Decreased p53 binding and inhibition	Cys77	[178]
Fas	Activated Fas/FasL signaling pathway	Increased sensitivity to Fas ligand	Increased cancer cell apoptosis	Cys304	[179]
MKP1	–	Increased phosphatase activity	Decreased radiation induced apoptosis	Cys258	[154]
TRAP1	Increased mitochondrial ROS production & permeability transition pore opening	Promoted proteasomal degradation	Increased cell death in GSNOR deficient cells (HCC)	Cys501	[33,180,181]

**Table 3 ijms-22-04600-t003:** S-Nitrosylation in anti-cancer therapy.

Drug	Molecular Signaling Changes	Biological Outcome	Model and Cell Type	Reference
**Reducing S-Nitrosylation**				
1400W, L-NAME, L-NMMA	iNOS inhibition, HIF-1α, and IRE1α/XBP1 impairment	Decreased cell growth and motility	TNBC, MDA-MB-231 and SUM159	[251]
L-NMMA+ Docetaxel	iNOS inhibition, ASK1 activation	Increased cytotoxicity in docetaxel-resistant cells	TNBC, SUM-159PT, MDA-MB-436, and MDA-MB-468	[257]
1400W	Rescues AGT depletion	Reduced DNA mutagenesis	HCC, Diethylnitrosamine (DEN) induced HCC in murine model	[258]
MDA-7/IL-24	Increased BCL-2 denitrosylation	Increased apoptosis	Pan cancer, melanoma A375, and renal carcinoma 7860	[260]
1400W	Increased OGG1 activity	Increased DNA-repair activity	Cholangiocarcinoma, KMBC	[24]
1400W, L-NIO	Inhibition of angiogenesis related genes	Decreased cell growth, migration, and angiogenesis	CRC; HT 29, and HCT 116	[217]
L-NAME	Inhibition of MAPK signaling	Decreased cell growth and survival	Breast cancer, LM-2, LM-3, LMM3, MDA-MB-231	[22]
**Increasing S-nitrosylation**				
SNP, GSNO	Increased ERK1/2 S-nitrosylation	Decreased cell growth	Glioma, U251 cells	
GSNO	Increased STAT3 S-nitrosylation	Decreased cell growth of chemo-resistant cells	Ovarian cancer. Ovarian cancer cell lines and HNSCC	[105,261]
GTN	cIAP S-nitrosylation	Increased apoptosis and cell death	Colon and breast cancer. SW480, CT26, MDA-MB-231, and EMT6, macrophages	[262]
JSK	Inhibition of ubiquitination	Decreased cell growth	Prostate cancer, LNCaP, and C4-2	[263]
NO-ASA and NO-naproxen	Increased NF-κB S-nitrosylation	Decreased cell growth	Colon cancer, HT-29 cells	[264]
NO-NSAID	Increased NF-κB and caspase-3 S-nitrosylation	Decreased cell growth	Pan-cancer	[265]
SNOC, GSNO, and DETA-NO	Increased Androgen receptor	Decreased cell growth	Prostate cancer, LNCaP, PC3, and 22Rv1 cells	[28]
SNP	Increased ERK1/2 S-nitrosylation	Increased apoptosis	Breast cancer, MCF-7 cells	[173]

## Data Availability

Not applicable.

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
