# Peer review of "S-Nitrosylation in Tumor Microenvironment"

_ijms, 2021, doi:10.3390/ijms22094600_

Round 1

Reviewer 1 Report

This is an excellent and informative review article. I thoroughly enjoyed reading it. S-nitrosylation is an important post-translational modification of proteins that is shown to be important in modulating the signaling activities of a cell, contributing to cancer. This article focusses on the tumor microenvironment and hence focusses on the role of S-nitrosylation in different cells that are present in the tumor microenvironment.

Following are some of the comments:

  1. It is a good idea to start with describing role of nitric oxide in cancer before starting with S-nitrosylation
  2. Is there evidence on whether specific oncogenic mutations have more or less degree of S-nitrosylation? If yes, then it would be great to put that data in a tabular form.
  3. There is evidence of profiling and association of s-nitrosylation in human pancreatic ductal adenocarcinoma pathogenesis. It would be great to add a section to highlight the importance of s-nitrosylation in different human cancers.
  4. Spell check line 110: denitrosylases
  5. Table 1 has proteins that are nitrosylated and their biological effects. Please add the molecular signaling changes upon nitrosylation (for example: activation of MAPK signaling pathway) that are reported which in turn caused the biological effects.

Author Response

Reviewer 1

Comments and Suggestions for Authors

This is an excellent and informative review article. I thoroughly enjoyed reading it. S-nitrosylation is an important post-translational modification of proteins that is shown to be important in modulating the signaling activities of a cell, contributing to cancer. This article focusses on the tumor microenvironment and hence focusses on the role of S-nitrosylation in different cells that are present in the tumor microenvironment.

Response: We appreciate the reviewer’s comment.

Following are some of the comments:

  1. It is a good idea to start with describing role of nitric oxide in cancer before starting with S-nitrosylation

Response: We appreciate and agree with the reviewer’s constructive suggestion. We have revised manuscript to include the section describing the role of nitric oxide in cancer (Section 2. Nitric Oxide (NO) Signaling) prior to the subsequent sections describing the role of S-nitrosylation in cancer.

  1. Is there evidence on whether specific oncogenic mutations have more or less degree of S-nitrosylation? If yes, then it would be great to put that data in a tabular form.

Response: Yes, there is a well–known case for p21Ras, where oncogenic mutations incorporate two additional S-nitrosylation sites to promote nucleotide exchange of the catalytic site. We have created a new section (4.1.3. S-Nitrosylation Influenced by Oncogenic Mutations) to discuss this aspect. We have also included oncogenic for p21Ras in the Table 1 that lists different S-nitrosylated proteins.

  1. There is evidence of profiling and association of s-nitrosylation in human pancreatic ductal adenocarcinoma pathogenesis. It would be great to add a section to highlight the importance of s-nitrosylation in different human cancers.

Response: We appreciate and agree with the reviewer’s constructive suggestion.  We have created a new section (4.1. S-Nitrosylation in Cancer) to discuss the roles of S-nitrosylation in different human cancers including pancreatic ductal adenocarcinoma.

  1. Spell check line 110: denitrosylases

Response: Thank you very much for pointing out the typo.  We have corrected the error.

  1. Table 1 has proteins that are nitrosylated and their biological effects. Please add the molecular signaling changes upon nitrosylation (for example: activation of MAPK signaling pathway) that are reported which in turn caused the biological effects.

Response: We appreciate and agree with the reviewer’s suggestion. We have revised Table 1 to include the molecular signaling directly influenced by S-nitrosylation.

Reviewer 2 Report

In this review, the authors discuss the role of S-nitrosylation in major constituents of the tumor microenvironment including whether this post-translational modification has a tumor promoting or suppressing effect in specific cases. Overall, the authors discuss a range of cell types in the tumor micro-environment and have presented the literature well.

Some minor comments -

1. please define all abbreviations before usage (for example, line 98 - GSNO, GAPDH; Fig 2 – TAMs, THs)

2. Line 128: Please use AMP Kinase or AMPK.

3. It would be nice to tabulate the discussion in section 3 – “S-Nitrosylation in Diseases”. A table listing major S-nitrosylation sites (or nitrosylated proteins) in different types of cancer (or sub types of cancer) would add value – similar to table 1 (currently).

4. It would be nice to add a section on bioinformatics analysis of nitrosylation sites. Since the targets of S-NO is varied, it would be interesting to see if the sites are conserved across different sub-types of cancer or even across different cancers.

5. Another aspect that is missing is discussion on normoxia vs hypoxia. Hypoxia is believed to enchance S-nitrosylation mediated responses. Are such responses important in cancer? This would be good to include.

Author Response

Reviewer 2

In this review, the authors discuss the role of S-nitrosylation in major constituents of the tumor microenvironment including whether this post-translational modification has a tumor promoting or suppressing effect in specific cases. Overall, the authors discuss a range of cell types in the tumor microenvironment and have presented the literature well.

Response: We appreciate the reviewer’s comment.

Some minor comments -

  1. please define all abbreviations before usage (for example, line 98 - GSNO, GAPDH; Fig 2 – TAMs, THs)

Response: We appreciate the reviewer’s comment.  We have gone through all the text and defined all the abbreviations before the first-time usage.

  1. Line 128: Please use AMP Kinase or AMPK.

Response: We appreciate the reviewer’s comment.  We have used AMPK throughout the manuscript.

  1. It would be nice to tabulate the discussion in section 3 – “S-Nitrosylation in Diseases”. A table listing major S-nitrosylation sites (or nitrosylated proteins) in different types of cancer (or sub types of cancer) would add value – similar to table 1 (currently).

Response: We appreciate the reviewer’s comment.  We have created a new section (4.1. S-Nitrosylation in Cancer) as well as a new table (renamed Table 1) to discuss the roles of S-nitrosylation in different types of cancer. 

  1. It would be nice to add a section on bioinformatics analysis of nitrosylation sites. Since the targets of S-NO is varied, it would be interesting to see if the sites are conserved across different sub-types of cancer or even across different cancers.

Response: We appreciate the reviewer’s comment.  We have created a new table (renamed Table 1) to describe bioinformatics analysis of S-nitrosylation sites in different cancers.

  1. Another aspect that is missing is discussion on normoxia vs hypoxia. Hypoxia is believed to enhance S-nitrosylation mediated responses. Are such responses important in cancer? This would be good to include.

Response: We appreciate and agree with the reviewer’s constructive suggestion. We have created a new section (4.1.2. S-Nitrosylation Influenced by hypoxia) to discuss the roles of S-nitrosylation in cancer under hypoxia.

Reviewer 3 Report

In this article, authors summarize the impact of endogenous nitric oxide level and S-Nitrosylation in some regulatory proteins associated with tumor development or suppression and finally authors discuss their effects in modulation of components in tumor microenvironment. This is well written and informative. 

Comments:

  1. Lines 169- 170: “The overall consequences of this protein modification depend on the context, cell type, target proteins, and the resultant molecular events”- How does context (of what?) or cell type variation influence pathogenesis of disease and cancer progression?
  2. Table 1: authors should include outcome of protein S-Nitrosylation (activation/ inactivation/localization or stability).
  3. Is there any cancer type specific signature of S-Nitrosylation? Whether the pattern indicates disease diagnosis or prognosis?
  4. If authors can provide a tabulated representation of “S–Nitrosylation in Anti–cancer Therapy”- indicating specific drug, its specific target (protein and tumor micro-environmental component) and its pre-clinical/ clinical observation.
  5. Authors should include the challenges in S-Nitrosylation mediated cancer therapy, specifically on non-specific effect and mode of specific target.

Author Response

Reviewer 3

In this article, authors summarize the impact of endogenous nitric oxide level and S-Nitrosylation in some regulatory proteins associated with tumor development or suppression and finally authors discuss their effects in modulation of components in tumor microenvironment. This is well written and informative. 

Response:  We appreciate the reviewer’s comment.

Comments:

  1. Lines 169- 170: “The overall consequences of this protein modification depend on the context, cell type, target proteins, and the resultant molecular events”- How does context (of what?) or cell type variation influence pathogenesis of disease and cancer progression?

Response: We appreciate the reviewer’s comment.  We have revised the whole section of 4. S–Nitrosylation in Diseases to give more concrete examples of dichotomous effects of S-nitrosylation depending on the context, cell type and target protein.

  1. Table 1: authors should include outcome of protein S-Nitrosylation (activation/ inactivation/localization or stability).

Response: We appreciate the reviewer’s comment.  We have revised the table (Table 2 in the revised manuscript) to include the direct consequence of S-nitrosylation on the target protein.

  1. Is there any cancer type specific signature of S-Nitrosylation? Whether the pattern indicates disease diagnosis or prognosis?

Response: We appreciate the reviewer’s comment.  We have created a new table (renamed Table 1) to describe S-nitrosylation in different cancer types.  Nevertheless, there has not been a comparative study on S-nitrosylation in different cancer types and stages. Thus, it is difficult, at this point, to determine that these S-nitrosylated proteins are unique to a particular type of cancer or associated with disease progression or prognosis.  We do agree that these aspects warrant further investigations.

  1. If authors can provide a tabulated representation of “S–Nitrosylation in Anti–cancer Therapy”- indicating specific drug, its specific target (protein and tumor micro-environmental component) and its pre-clinical/ clinical observation.

Response: We appreciate the reviewer’s constructive suggestion.  We have created a new table (Table 3) to describe the use of S-nitrosylation in anti-cancer therapy.

  1. Authors should include the challenges in S-Nitrosylation mediated cancer therapy, specifically on non-specific effect and mode of specific target.

Response: We appreciate the reviewer’s constructive suggestion.  We have included our comment on the challenges in S-nitrosylation-mediated cancer therapy.

Round 2

Reviewer 3 Report

Sharma et al have revised the manuscript entitled "S–Nitrosylation in Tumor Microenvironment" properly.